# A Biomimetic Flexible Sliding Suction Cup Suitable for Curved Surfaces

**DOI:** 10.3390/biomimetics10030137

**Published:** 2025-02-24

**Authors:** Enhua Cui, Xiangcong Zhou, Yanqiang Liu, Jixiao Xue, Siyuan Xiong, Deyuan Zhang

**Affiliations:** School of Mechanical Engineering & Automation, Beihang University (Beijing University of Aeronautics and Astronautics), Beijing 100191, China; by2207105@buaa.edu.cn (E.C.); 18810950892@163.com (X.Z.); 18811709882@163.com (J.X.); xxycjwsy0930@163.com (S.X.)

**Keywords:** wall-crawling robot, biomimetic sliding suction cup, curved surface crawling, friction-reduced structure, bionic flow channel structure

## Abstract

The sliding suction robots designed for wall-climbing functions could have accuracy defects due to suction cup sealing, friction interference, and surface adaptability. Hence, this work develops a biomimetic, flexible, sliding suction cup suitable for crawling on curved surfaces. Inspired by the hypostomus plecostomus’s mouth, we designed a biomimetic low-contact force flow channel structure and a matrix of friction-reducing protrusions along the lip edge of the sliding suction cup. This design reduces frictional resistance on the sliding interface and the flexible nature of the suction cup, allowing it to be used on curved or vertical surfaces of different materials. Several simulation-based optimization analyses and experimental tests are conducted on the biomimetic low-contact force flow channel structure, and various structural design principles are explored for achieving high adhesion and low-contact force. Additionally, a friction reduction model for the matrix structure is designed to verify the effects of parameters such as load, protrusion size, and quantity on the friction coefficient of the matrix structure surface through friction tests. The sliding suction cup prototype presents an average crawling speed of about 0.4 m/s on a horizontal plane and 0.7 m/s for crawling on vertical walls and the inner surface of a cylindrical rail.

## 1. Introduction

Manufacturing continuous motion on aircraft skin, pipelines, buildings, ships, and other spatial curved surfaces involves the challenge of adhesion on vertical walls and inverted top surfaces [1,2,3]. This challenge is addressed through magnetic adhesion [4,5,6], vacuum adhesion [7,8,9,10], electrostatic adhesion [11,12], adhesive bonding [13,14], and mechanical locking [15,16]. Vacuum adhesion has a broad application range, but the contradiction between suction cup sealing and motion friction and the suction cup surface adaptability issue [17] has remained unresolved.

Although combining rigid suction cups with flexible seals has been widely employed, it is unsuitable for highly curved motion surfaces, as many flexible lip suction cups have high friction after adhesion and cannot move effectively [18]. Existing interface friction reduction techniques primarily reduce pressure [19] and lower the friction coefficient. Reducing pressure by providing support from air or liquid at the contact interface is a common method for reducing friction in ultra-precision machine tools [20,21,22] and precision instruments [23,24,25]. For instance, Fan et al. developed a multi-microhole air bearing and linear sliding system, which utilizes modern MEMS technology to create micro-diameter holes to enhance airflow stability, resulting in good pressure capacity and stability [26]. Besides, Chen et al. analyzed the pressure distribution, flow rate, and load-bearing stiffness of the gas film of static pressure gas bearings under rarefaction [27]. They indicated that gas supply pressure affects the optimal load-bearing capacity of static-pressure gas bearings. Jonathon et al. verified that an asymmetric deep contour surface can generate loads through viscous effects in Newtonian fluids, thus reducing friction [28]. Besides, interface support can reduce pressure and friction, and thus, an interface microstructure design based on biomimetic principles can improve interface friction performance [29,30,31]. An array of convex protrusions is a typical micro-structure for friction reduction [32]. Ren et al. researched the anti-adhesive properties of beetle surfaces and developed biomimetic plows with convex-micro-structure interfaces, which reduced plowing resistance by 15–18% compared to regular plows. Similarly, Soni et al. [33] designed template plows using polyethylene protrusions based on beetle surface features. They discussed and analyzed the impact of protrusion distribution and shape on reducing plow resistance and concluded that, when the non-dimensional height-to-diameter ratio of protrusions is less than 0.5, it is more conducive to reducing plowing resistance [34]. Nevertheless, most of these studies reduce pressure through external forces or media, and most protrusion friction reduction is determined through experimental exploration of structural parameters. Hence, simple methods and scientifically designed theories for reducing pressure and friction coefficients in sliding suction cup interfaces are lacking.

The hypostomus plecostomus (pleco) is a typical organism in nature that has micro-protrusion structures that have been proven to reduce interfacial friction [35,36]. In still currents, the pleco can be observed to slide forward slowly while clinging to irregular surfaces like rocks [35,36,37,38]. This adsorption-sliding movement has similar motion characteristics to a sliding suction cup, which provides us with bionic inspiration for further research. Through our observation, the pleco’s lower lip surface is covered with protrusion structures of approximately 200 μm, characterized by small, dense edges and larger sparser interiors. Figure 1 shows that the lip grooves are located on both sides of the suction cup’s mouth, connecting the internal oral cavity with the external environment, allowing it to breathe while adhering to surfaces and regulating the adhesion force of the suction cup. Tan et al. analyzed the balanced mechanism of the tight surface adhesion and fast sliding of the rock-climbing fish with similar micro-protrusion structures, and proved that micro-protrusion hydrodynamic interaction and sealing suction cups work cohesively to contribute to low friction and high pull-off-force resistance and can therefore slide rapidly while clinging to the surface [39]. Besides, Zou et al. found through an experiment that the effect of the micro-protrusion structure to enhance adsorption-sliding movement is most likely related to the accumulation of micro-bubbles in the liquid [40]. However, research on the anti-friction effect of the pleco lip’s micro-protrusion structure mainly focuses on the solid-liquid interface, where the capillary force and Stefan force of the liquid play an important role in the entire movement process. To further expand the application of the sliding suction cup, the anti-friction effect and gas flow change in the sliding suction cup with protrusion structures on solid surfaces need to be further studied.

This work drew inspiration from a pleco’s mouth to design a biomimetic low-contact force flow channel structure on the lip edge of the sliding suction cup. The pressure through the channels on the lip edge of the suction cup is reduced by relieving positive pressure, and a biomimetic micro-protrusion structure surface, which is similar to the pleco’s mouth, is also utilized for friction reduction of the base plate. This approach ensures that the reduced pressure suction cup has sufficient adhesion during crawling while reducing the contact pressure between the lip edge and the wall. Additionally, a matrix design on the lip edge surface is incorporated to reduce the interface friction coefficient, thereby lowering the friction resistance of the sliding interface. This crawling solution applies to flat and curved surfaces of various materials.

## 2. Materials and Methods

### 2.1. Design of Biomimetic Flexible Suction Cup

#### 2.1.1. Biomimetic Flexible Suction Cup System Design

Inspired by the adhesion and crawling movement of the pleco, we propose a biomimetic flexible lip edge design that incorporates a protrusion structure and pressure-relief holes, aiming to achieve a low-pressure friction reduction effect. Figure 2 depicts the physical design and production of the biomimetic suction cup. The biomimetic suction cup employs a three-wheel drive design with a total mass of approximately 650 g (excluding batteries and spraying devices). The inner diameter of the reduced pressure chamber is 160 mm, and when placed flat on a surface, the maximum diameter of the lip edge is 320 mm.

The driving wheels were created using biomimetic high-friction rubber covering a PLA hub, and the skin of the wheel had an equivalent friction coefficient greater than 1. A brushless DC gear motor (RoboMaster M2006 P36, by DJI, ShenZhen, China) was used. This motor has a small volume, high power density, and high output speed. The reduction ratio of the gearbox was 36:1. Considering the position of the center of gravity, a pair of nylon umbrella gears with a gear ratio of 1:1 was used for transmission. The diameter of the driving wheels (D) was 30 mm with a width (W) of 15 mm. If the limiting speed of the driving platform is v = 0.4 m/s, the required frictional force for wheel propulsion, FD, was 50 N, which is twice the weight of the designed platform. Therefore, the motor speed, torque, and power requirements are n = 255 rpm, T = 0.75 N·m, and P = 20 W, respectively. Combined with the motor load characteristic curve and the calculated results, the selected motor meets the requirements of the prototype. The motor was controlled in an open-loop speed controller using a C610 (by DJI, ShenZhen, China), powered by a 24 V DC power supply in the experiment.

#### 2.1.2. Design of Biomimetic Protrusion Structures

The pressure relief reduces the contact pressure at the interface, while the protrusion structure reduces the contact area, lowering the friction coefficient. Biomimetic protrusion structures relieve pressure and reduce the friction coefficient by minimizing the actual contact area. Indeed, the contact between the flexible protrusions and the surface of the substrate is linked with the interaction between two elastic bodies, which can be approximated as the contact between an elastic sphere and a rigid, smooth plane. Under the action of load W, the radius of the protrusion contact surface changes and results in a deformation δ of the normal displacement, and the effective contact area radius is denoted as a. In this context, the equivalent elastic modulus E′ and the equivalent radius R must be corrected as follows:(1)$$\frac{1}{E\prime }=\frac{1-{\nu_{1}}^{2}}{E_{1}}+\frac{1\;-{\nu_{2}}^{2}}{E_{2}}$$(2)$$\frac{1}{R}=\frac{1}{R_{1}}+\frac{1}{R_{2}}$$
where E_1_ and E_2_ represent the elastic modulus of the protrusion and the coating, respectively, ν_1_ and ν_2_ represent their Poisson’s ratio, respectively, and R1 and R2 are their respective radio, elastic mechanics analysis reveals that:(3)$$\delta ={\left(\frac{9W^{2}}{16E\prime^{2}R}\right)\;}^{\frac{1}{3}}$$(4)$$a={\left(\;\frac{3\mathrm{WR}}{4E\prime }\right)\;}^{\frac{1}{3}}$$(5)$$W=\frac{4}{3}E\prime R^{\frac{1}{2}}\delta^{\frac{3}{2}}$$

The actual contact area of a single protrusion can be derived from the above relationships as follows:(6)$$A=\pi a^{2}=\pi R\delta$$

When the lip surface comprises n equally sized protrusions, and assuming each protrusion bears a load of W, the total load W_T_ is expressed as:(7)$$W_{T}=\mathrm{nW}=\frac{4}{3}\mathrm{nE}\prime R^{\frac{1}{2}}\delta^{\frac{3}{2}}$$

The actual total contact area A_T_ between the lip surface and the coating is:(8)$$A_{T}=\mathrm{nA}=n\pi R\delta$$

By substituting Equation (7) into Equation (8),(9)$$A_{T}={\left(\frac{3R}{4E\prime }\right)}^{\frac{2}{3}}\pi n^{\frac{1}{3}}{W_{T}}^{\frac{2}{3}}$$

Thus, we have derived the relationship between the normal deformation of the biomimetic flexible micro-protrusion array surface and the coating, the total actual contact area, geometric dimensions, protrusion density, and the load.

Since the silicone rubber is naturally soft and easily deformable when contacting the coating surface, a larger actual contact area is generated. The initial friction experiments revealed that the equivalent coefficient of friction decreases as the load increases. Therefore, analyzing its friction characteristics using the molecular-mechanical theory is most appropriate. For elastic contact friction, the actual contact area is linearly proportional to the normal load to the power of two-thirds, therefore, the equivalent friction coefficient decreases with the increase in load. According to the molecular-mechanical theory, the equivalent coefficient of friction f is given by:(10)$$f=\frac{\alpha A}{W}+\beta$$
where the physical and mechanical properties of the frictional surfaces determine α and β represents the actual coefficient of friction, and α is a constant. From the elastic contact theory, under the constraint of the elastic contact range, and by combining Equations (9) and (10), the equivalent coefficient of friction is given by:(11)$$\mu =\pi \alpha {\left(\frac{3R}{4E^{\prime }}\right)}^{\frac{2}{3}}{\left(\frac{n}{W_{T}}\right)}^{\frac{1}{3}}+\beta \;$$

Hence, when designing an array of convex structures for elastic friction reduction, the parameters to be satisfied are the curvature radius, material hardness, total load, and the number of convex features. Additionally, the relationship between these design parameters must ensure that only the convex features experience elastic contact, resulting in an equivalent coefficient of friction. The equation is satisfied when the equivalent coefficient of friction is minimized concerning the design parameters.

To validate the correctness of the biomimetic array convex surface friction reduction theoretical model, we conducted experimental tests to verify the friction reduction performance of the biomimetic interface. We aimed to understand the trend and characteristics of the equivalent friction coefficient under varying key parameters. The experimental investigation was conducted using a controlled variable approach, where the convex size, density, and normal load were the experimental variables, to explore the friction characteristics of the biomimetic flexible array convex surface. Figure 3a illustrates the parameter design of the biomimetic array convex sample used in the experiments. Figure 3b depicts the experimental testing platform for measuring frictional forces on the biomimetic array convex surface, where the contact material is a layer of thermoplastic elastomer. Figure 3c presents the equivalent friction coefficient variations under different loads, convex sizes, and densities. The results highlight that excessive load or too small a convex size leads to structural deformation failure. As the load W increases, the equivalent friction coefficient f gradually decreases, which correlates linearly with W^(−1/3)^. Under the same load, f first decreases and then increases as the lattice diameter increases. At low loads, the specimen’s equivalent f is linearly related to D^(2/3)^. For the same load, the equivalent f increases gradually as the number of convex features increases, and the growth rate becomes slower. The relationship between f and the number of convex features approximately follows n^(1/3)^, showing a linear correlation. The above experiments reveal that the structure’s convex, flexible array has a friction reduction pattern that closely matches the theoretical analysis.

### 2.2. Manufacturing of Biomimetic Flexible Suction Cups

For molding the suction cup, we chose a conventional silicone rubber mold with relatively high hardness, ensuring a deformation adhesion performance of the lip edge to curved surfaces and structural strength and wear resistance. Moreover, this material is cost-effective. The manufacturing process of the suction cup primarily involves casting, molding, 3D printing technologies, and other techniques. The detailed process for fabricating the biomimetic flexible sliding suction cup is as follows (Figure 4):

The mold components were manufactured using 3D printing. Specifically, the biomimetic structural elements were created using the Anycubic Photon M3 UV resin 3D printer with a printing accuracy of up to 0.01 mm, meeting the manufacturing requirements for biomimetic features on the surface of the suction cup. A suitable amount of demolding agent or petroleum jelly was applied to the mold’s inner surface to facilitate easy removal from the mold. It should be noted that any excess demolding agent should be carefully removed from the surface using a high-pressure air gun to ensure even distribution and avoid any adverse effects on the fabrication of the protrusion array.

The silicone and curing agent were mixed at a 100:2 mass ratio. After stirring, a vacuum chamber removed the air bubbles from the silicone mixture, which was poured into the mold. Due to the mold’s depth and the possibility of reintroducing bubbles during pouring, a two-step casting approach was utilized. Initially, a small quantity of silicone was poured that did not cover the entire base of the mold. Vacuum deaeration was performed to eliminate any newly introduced air bubbles. This step ensured the production quality of the flexible lip structure. Subsequently, more silicone was poured to fill the cavity before the vacuum deaeration again to eliminate bubbles generated during the second pouring. The vacuum deaeration was continued until all bubbles were removed. The poured silicone was left to cure at room temperature (approximately 25 °C) for 4 h. After the curing, the mold was disassembled, and the flexible sliding suction cup was released.

## 3. Results and Discussion

### 3.1. Biomimetic Sliding Suction Cup Crawling Experiment

The biomimetic sliding suction cup robot is able to crawl on ceiling planes (Figure 5a), inner cylindrical surfaces (Figure 5b), and vertical planes (Figure 5c,d). The suction cup starts from rest and is driven by a constant torque. The prototype achieves an average crawling speed of approximately 0.4 m/s on vertical planes while hanging upside down and approximately 0.7 m/s on vertical walls and inside cylindrical curved surfaces.

The crawling tests (Figure 5) revealed that during the top-surface crawling process, the adhesive force generated by the suction cup overcomes gravity and provides sufficient normal pressure to the driving wheel, enabling stable crawling. The biomimetic flexible lip edge remains tightly attached to the wall surface without rolling or other occurrences that could lead to gas leakage, demonstrating excellent adhesion stability.

During the top crawling process, under the influence of the adhesive force, the driving wheel generated enough static friction to overcome its weight and the frictional resistance of the lip edge, achieving stable crawling at a speed similar to that of top-surface crawling. The biomimetic sliding suction cup exhibited adequate adhesion and crawling in the side-crawling process. Due to balancing the suction cup’s gravitational force through the frictional forces at the lip edge and the tire’s lateral friction, the sliding suction cup achieved a faster driving speed under the same driving power, approximately 0.6 m/s. Furthermore, it maintained good adhesion stability during fast crawling. In the cylindrical surface crawling test, the suction cup could advance along the curved surface in a spiral manner. Within 1 s, it effectively completed stable crawling on the lower half of the curved surface, with an average crawling speed of no less than 0.66 m/s. The suction cup’s lip edge maintained excellent contact with the curved surface throughout the crawling process, showcasing its adaptability. In conclusion, the biomimetic sliding suction cup demonstrates impressive abilities for both planar and curved surface adhesion and crawling.

### 3.2. Flow Channel Structure Optimization Design and Experiment

Appropriately reducing the positive pressure at the lip edge of the suction cup resolves the conflict between sealing and motion friction. When the suction cup adheres to a surface, the relief holes and protrusion structures form gaps, constituting a low-contact-force flow channel structure. During the sliding suction cup’s adhesion to the surface, a controlled amount of air enters the lip edge contact area through the relief holes, reducing the contact interface’s vacuum level and lowering the contact pressure and friction force. However, introducing flow channel structures can compromise the sealing performance of the suction cup. Therefore, designing and optimizing the flow channel structure is required to ensure that the sliding suction cup has both low contact force and high adhesion.

We designed and simulated the flow field at the sealing lip interface using the CFD module of COMSOL Multiphysics (6.3). During this process, we assumed that external gas enters the contact area only through the relief holes when the suction cup is in adhesion mode. Figure 6a–c illustrate the three different interface geometrical models designed with varying relief hole diameters. Besides, the flow field and pressure distribution at the contact interface with different relief hole sizes were computed, where a 1/4 sector was chosen for calculation, including the relief holes and the protrusion features. The complete contact area was obtained through symmetry treatment. The relief hole diameters for the three models were 0.2, 0.4, and 0.6 mm, respectively, while the other parameters remained constant. Specifically, the cylinder diameter was set to 0.3 mm, and the gap height at the contact interface was 0.12 mm. The inlet air pressure was set to atmospheric pressure, and the outlet air pressure was set to 10 kPa below ambient. An auxiliary scan was used to gradually decrease the pressure from 0 to the specified pressure to improve the convergence of the calculations. Finally, the simulation type was set to steady-state, and the calculations were performed. The flow field distribution maps revealed that external air flows into the contact interface through the relief holes. Additionally, the flow velocity was higher in the relief hole area, and guided by the gaps formed by the lattice structure, the gas diffused radially towards the interface and ultimately entered the low-pressure region within the chamber.

The pressure distribution highlighted that due to the influence of the gas entering through the relief holes, the vacuum level in the outer ring of the contact interface decreased significantly, and the area with higher flow velocity, especially near the relief holes, exhibited lower pressure. As the relief hole diameter increased, the overall average vacuum pressure of the contact interface gradually decreased, indicating a reduction in the contact pressure and a more pronounced relief effect. Figure 6d depicts the variations in gas leakage and average contact pressure with increasing outlet relative negative pressure for 0.2 mm to 1 mm relief hole diameters. Figure 6d infers that the gas leakage increases gradually with the increase in relief hole diameter, and the rate of increase remains relatively constant. When the relief hole diameter increases from 0.2 mm to 1 mm, the unit pressure relief volume increases from 1.8 × 10^−5^ to 8.01 × 10^−5^ m^3^/s, and the maximum rate appears when the relief hole diameter increases to 0.6 mm, of 445%. The average relative negative pressure decreases linearly with the increase in relief hole diameter. At a central pressure of 10 kPa below ambient, the relief hole with a diameter of 0.2 mm reduces the average pressure at the contact interface to 9 kPa below ambient, with a relief magnitude of 1 kPa. On the other hand, the 1-mm relief hole decreases the average pressure at the contact interface to 3231 Pa below ambient, with a relief magnitude of 6769 Pa, representing a 576.9% increase in relief magnitude. Therefore, adjusting the relief hole diameter controls the relief effect and leads to substantial variations in gas leakage. By examining the trend of relief magnitude produced per unit gas leakage concerning relief hole diameter, we observe that the unit relief magnitude increases first and then decreases as the relief hole diameter increases, which maximizes when the relief hole diameter is 0.6 mm.

The interface gap formed by the protrusion structure also affects the interface sealing and pressure relief performance. Hence, to optimize the design of the lip interface gap parameters of the sliding suction cup, we designed contact models with height gaps of 0.08 mm, 0.12 mm, and 0.16 mm to explore the effect of height gap variation on the generated interface flow field. The relief hole diameter was set to 0.4 mm, while the other parameters were the ones mentioned above. Figure 7a–c show the flow and pressure field contour maps, highlighting that the circumferential flow distribution of gas in the interface gap is generally the same. As the gap height increases, the gas flow velocity on the contact interface gradually increases, especially in the high-speed region near the relief hole. This is due to the inhibitory effect of viscous resistance on the fluid when the height gap is too small. Additionally, the pressure distribution contour maps reveal that, as the height gap increases, the pressure relief effect of the interface gradually decreases, and the low-pressure area near the relief hole gradually increases. This is mainly due to the flow velocity increase. Figure 7d represents the gas leakage rate and relative pressure of the interface under three different height gaps during steady-state conditions. Figure 7d reveals that the gas leakage rate is sensitive to changes in the interface height gap, increasing linearly with height. When the height gap increases from 0.08 mm to 0.16 mm, the gas leakage rate increases by approximately 35.7%. Besides, the unit pressure relief volume gradually decreases as the height gap increases. The 0.08 mm height gap ensures the minimum gas leakage rate while having the best pressure relief effect. The average interface pressure under the three different height gap sizes is between 6.8 and 7.2 kPa below ambient, and the difference in pressure relief between them is less than 10%, indicating that the pressure relief effect does not differ significantly. When only investigating the gas pressure relief, a smaller interface height gap is better. However, an excessively small gap will increase the contact area between the flexible lip and the wall, thereby increasing intermolecular forces and reducing the pressure relief effect.

Inspired by the varying sizes of protrusions on the lower jaw of plecos, we investigated the flow field distribution and pressure relief performance under different-sized protrusions, leading to different height gradient gaps, as shown in Figure 8, represented according to the gas flow direction. In Figure 8a, the height gap increases from 0.08 mm to 0.16 mm, indicated as 0.08–0.16 mm, while in Figure 8b, it decreases from 0.16 mm to 0.08 mm. The other parameters are consistent with the previous discussion. Comparing the two models’ flow field cloud maps and pressure field maps reveals that the 0.16–0.08 mm wedge gap configuration has increased flow velocity compared to the 0.08–0.16 mm configuration, and its pressure relief capacity is significantly enhanced. This is due to the 0.16–0.08 mm wedge gap configuration causing the converging effect on the interface airflow, generating a vertical dynamic pressure component, thus improving the pressure relief effect. Figure 8c compares the gas leakage rate and average interface pressure of the two gradient gaps, contrasting with the uniform 0.12 mm height gap of equal volume. The results suggest that the wedge-shaped gap increases the gas leakage rate, with the converging wedge-shaped gap having the highest gas leakage rate. Compared to the 0.12 mm uniform gap interface, the gas leakage increases for the 0.16–0.08 mm gap, and 0.08–0.16 mm gap interfaces are 24.7% and 10.0%, respectively. The average pressure map shows that the pressure relief effect of the converging wedge-shaped gap has been significantly improved, accounting for 19% of the pressure relief difference between the set center relative negative pressure zone and the outside air pressure for the 0.12 mm uniform gap interface. The diverging wedge-shaped gap exhibits average pressure similar to the uniform height gap. Figure 8c highlights that the 0.16–0.08 mm gap configuration has the maximum unit pressure relief rate, exhibiting superior pressure relief performance. Therefore, while ensuring complete sealing of the lip boundary, the pressure relief performance of the converging wedge-shaped gap is better.

We experimentally evaluated the effects of the biomimetic vent holes and convex array structure-induced interface gaps on the pressure relief and adhesion performance of the flexible lip edge of the sliding suction cup. Figure 9a illustrates the experimental specimen of the biomimetic sliding suction cup and the overall parameter design. In contrast, Figure 9b presents the adhesion force testing platform for the sliding suction cup. The experimental setup primarily includes a vacuum pump, pressure regulator, gas flowmeter, tension tester, sliding suction cup fixture platform, and other components. The sliding suction cup fixture platform simulates the adhesion force state under the crawling platform’s conditions. When placed flat on a surface, the lip edge of the suction cup contacts the four wheels, ensuring that the lip edge provides a sealing function. After applying relative negative pressure, only the lip edge of the sliding suction cup maintains contact with the wall while sliding. At this point, when removing the effect of the vehicle’s weight, the force measured by the load cell in the experimental setup reflects the force acting on the supporting wheels on the ground. This force is beneficial for adhesion and provides friction force for crawling. Figure 9c depicts the testing system for measuring the contact force between the lip edge and the wall when the sliding suction cup adheres. The adhesion force testing system uses the gas supply, delivery, and pressure control system. Four resistive thin-film pressure sensors with a thickness of 0.05 mm and a diameter of 1 mm are attached to the aluminum plate in a circular distribution for contact force measurement of the lip edge and the wall. During testing, the sliding suction cup fixture platform simulates the adhesion force state under the crawling platform’s conditions. The thin-film sensors are placed between the sealing lip edge and the wall. The contact force measured by the sensors in the contact area represents the contact force exerted by the sliding suction cup’s sealing lip edge. The impact of different parameters of the biomimetic structural design on the lip edge’s contact force is determined by neglecting the sealing interface’s flow field and mechanical changes caused by the presence of the sensors. This enables us to identify the influence trend of various biomimetic structural designs on the lip edge’s contact force.

Figure 9d highlights the impact of different pump pressure conditions on the performance of the biomimetic sliding suction cup with varying vent hole diameters. Four groups of sliding suction cups had vent hole diameters of 0 mm (control group, no vent holes), 0.6 mm, 0.8 mm, and 1 mm, while the other parameters remained constant. Each group of suction cups had four vent holes distributed uniformly around the circumference with a distance of 4.5 mm from the lip edge. The convex structures had a diameter of 0.5 mm, and the vent holes were interspersed with the convex structures. As the vent hole diameter increased, the gas leakage rate gradually rose, decreasing the suction cup’s adhesion and contact forces. The ratio of adhesion force to contact force for the sliding suction cup also exhibited a trend of initially increasing and decreasing with vent hole diameter. The sliding suction cup with a vent hole diameter of 0.6 mm had the highest adhesion-to-contact force ratio. Figure 9e presents the test results of the biomimetic suction cup under different height gap conditions. These tests included varying convex diameters of 0 mm, 0.3 mm, 0.5 mm, 0.3 mm–0.5 mm, and 0.5 mm–0.3 mm. The results infer that the group with 0.5 mm diameter equal-sized convex structures exhibited the highest gas leakage rate. Besides, the group with a converging height gap of 0.3–0.5 mm exhibited a similar but slightly decreased gas leakage rate compared to the 0.5 mm group. The group with a diverging height gap of 0.5–0.3 mm also had a gas leakage rate similar to the 0.3 mm group. This indicates that the diameter of the convex structures at the lip’s edge significantly impacts gas leakage. Excessively large convex diameters led to a significant gap between the lip edge and the wall, reducing the sealing effect and allowing a large amount of air to enter. Although the larger convex diameters increased gas leakage, the difference in adhesion force was insignificant compared to the situation without convex structures. Thus, convex structures effectively reduced the contact force on the lip edge, providing good pressure relief. Additionally, larger convex diameters did not show a clear advantage in the gas leakage rate because the gas flow can be approximated as incompressible, reducing the pressure relief effect in the contact area. Convex diameters of 0.3 mm and the group with a converging 0.5–0.3 mm height gap had a higher adhesion-to-contact force ratio, with the 0.5–0.3 mm group having slightly higher crawling and adhesion performance than the 0.3 mm group. In conclusion, an appropriate convex diameter can balance the suction cup’s gas leakage and adhesion force. Choosing the correct convex diameter can prevent excessive gas leakage, reduce adhesion forces, and minimize the contact force on the lip edge, achieving good adhesion and crawling performance.

### 3.3. Biomimetic Flexible Lip Edge Anti-Curling Design Analysis

In the sliding experiment of the biomimetic suction robot, the lip edge of the moving suction cup could flip over sometimes due to friction force, which causes serious leakage and disables the AS movement. In view of this phenomenon, an anti-curling design analysis is performed to reveal the sections of the lip edge with insufficient rigidity to curl under the combined action of the driving force and contact friction, leading to gas leakage and preventing the sliding suction cup from adhering to the wall. Figure 10a illustrates the force analysis of the lip edge’s critical section at the suction cup’s front end during motion. The external forces acting on the lip edge primarily include the contact force and frictional force at the wall-contacting end, the balancing driving force, supporting force, and equilibrium torque at the suction cup’s fixed end, as well as the distributed pressure caused by the pressure difference between the internal and external pressure chambers of the suction cup. Under the influence of external forces, the flexible lip edge undergoes an S-shaped deformation. The wall-contacting end protrudes inward toward the suction cup, while the end closer to the fixed part of the suction cup protrudes outward. This configuration effectively transmits the driving force to balance the frictional resistance at the wall-contacting end. The angle between the deformation slope of the S-shaped deformation at the point of curvature direction change and the direction of frictional force is defined as the deformation angle of the lip edge. In a safe state, the lip edge deformation angle is acute, at which point the principal stress at the point of curvature direction change has a component opposite to the direction of frictional force. When the lip edge deformation forms a right angle, the maximum driving force transmitted by the lip edge can be achieved. If this still does not meet the requirements for balancing frictional force, the lip edge may undergo excessive deformation under tangential external forces, leading to lip edge curling and gas leakage.

Figure 10b employs finite element analysis to vertically calculate the force-induced deformation of the lip edge cross-section at the front end of the suction cup. This validates the rationality of the curling analysis of the lip edge and provides theoretical guidance for the design of anti-curling thickness in the sliding suction cup. The material of the lip edge is set to the silicone parameters mentioned in the following chapter, with a thickness of 2 mm and a length of 8 mm. The initial contact point between the lip edge and the wall is fixed, and an external load of 1 KPa is applied to the outer side of the lip edge to simulate the simplified, reduced-pressure environment. The upper-end face of the lip edge undergoes only horizontal displacement and is subjected to a rightward boundary load of 20 N, 30 N, and 40 N. The steady-state deformation of the lip edge cross-section is calculated, revealing that under the effect of external forces, the deformation of the lip edge cross-section indeed forms an “S” shape. As the load increases, the displacement of lip edge deformation continuously grows, and the tangent slope at the point of curvature transition increases. When the load reaches 40 N, the deformation angle is approximately 90°, reaching the near-curling state. Therefore, the calculated force-deformation results of the lip edge align with the theoretical analysis. Figure 10c analyzes the deformation of two lip edges with different shapes but the same cross-sectional area. Under the same load, the lip edge with varying thickness has a smoother tangent slope at the curvature transition point compared to the uniform-thickness lip edge. The deformation angle of the lip edge is approximately 79°, and the maximum deformation displacement is 5.88 mm, which is lower than the 7.16 mm of the uniform-thickness lip edge. Consequently, the lip edge with varying thickness exhibits superior anti-curling performance.

## 4. Conclusions

This work presents a curved surface crawling robot prototype using a biomimetic sliding suction cup for negative-pressure adhesion. In response to the contradiction between suction cup sealing and motion friction, as well as the adaptability issue of the suction cup surface, this work designs and manufactures a low-contact-force flow channel structure and lattice structure for a biomimetic suction cup using the mouth of the pleco as a template. Inspired by the lip protrusion structure and the two lateral lip grooves of the pleco mouth, we propose a biomimetic sliding suction cup crawling system design scheme. This design aims to reduce the contact pressure and actual contact area (friction coefficient) to minimize sliding friction resistance. Numerical calculations and experimental optimization determined the impact of channel structure parameters on reducing the interface contact pressure. It is concluded that the sliding suction cup achieves the highest suction-pressure ratio with a vent hole diameter of 0.6 mm and a protrusion diameter of 0.3 mm.

The intrinsic relationship between lattice structure parameters and the equivalent friction coefficient is analyzed using elastic contact theory and the friction binomial law. Friction tests were conducted to verify the correctness of the theoretical derivation, which demonstrated that the equivalent friction coefficient improves gradually as the protrusion diameter decreases. Still, the ability to maintain the protrusion gap diminishes, creating new contact points. From the friction experiments, the optimal protrusion design parameters and minimum friction coefficients were determined for different loads. The manufacturing process for the biomimetic flexible suction cup was explored, and the biomimetic sliding suction cup system prototype was successfully fabricated. This prototype achieved a sliding speed of up to 0.66 m/s under a low equivalent friction coefficient.

Our future work will explore wear-resistant lattice structure design solutions and manufacturing processes. The dynamic characteristics of crawling on curved surfaces under load for the suction cup and intelligent motion control strategies will also be studied, aiming to achieve better performance in terms of sliding and crawling capabilities on various surfaces.

## Figures and Tables

**Figure 1 biomimetics-10-00137-f001:**
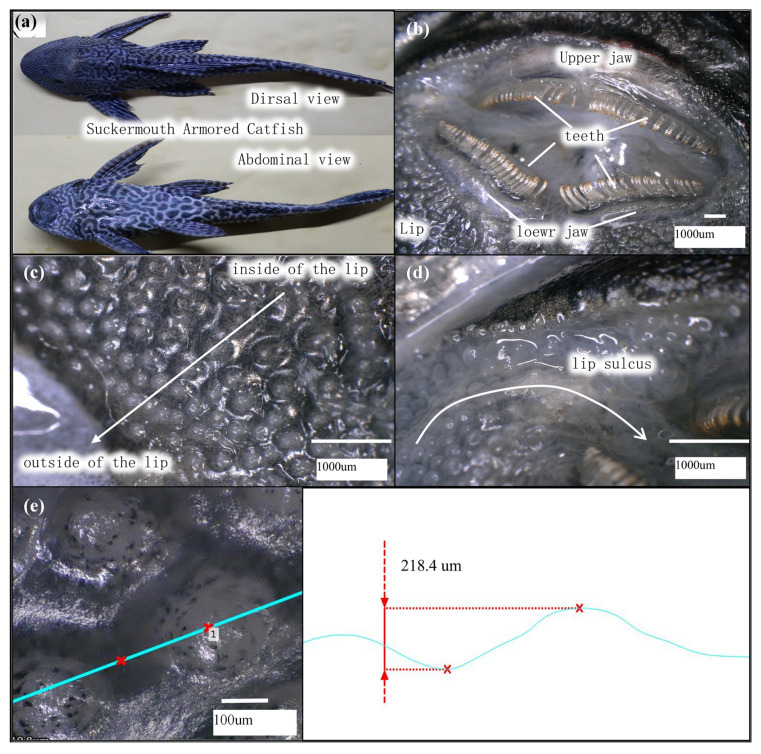
Observation of characteristics of the pleco adhesion chamber. (**a**) Dorsal and abdominal views of pleco. (**b**) Teeth and oral cavity structure of pleco. (**c**) Protrusion array structure on pleco’s lips. (**d**) Lip sulcus structure in pleco’s mouth. (**e**) Height difference between the top and the sulcus of protrusion structure.

**Figure 2 biomimetics-10-00137-f002:**
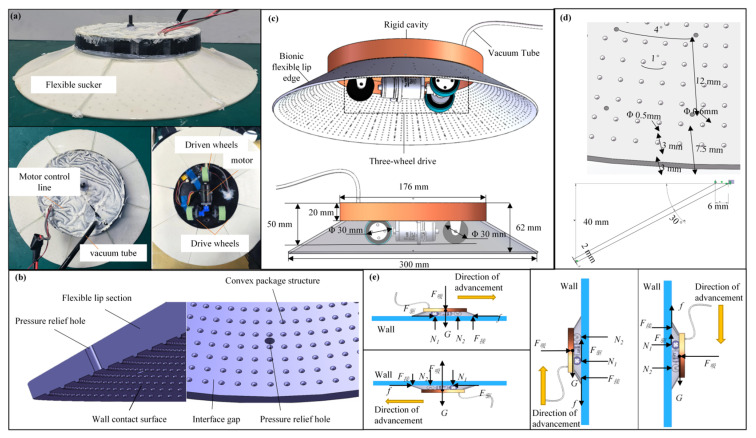
Biomimetic design, manufacturing, and force analysis of sliding suction cups. (**a**) Biomimetic sliding suction cup system; (**b**) biomimetic flexible lip edge design; (**c**) overall structural design of the sliding suction cup; (**d**) protrusion structural design of biomimetic flexible lip edge; (**e**) force analysis of uniform sliding on the ground, ceiling, and vertical surface. N_1_ and N_2_ are the pressure of the driving wheel and the driven wheel of the wall-climbing driving platform, respectively, F_S_ is the adsorption force of the suction cup chamber, F_D_ is the static friction force on the wall when the driving wheel is rolling, F_C_ is the pressure of the bionic flexible lip contacting the wall, f is the friction resistance, and G is the gravity force of the driving platform.

**Figure 3 biomimetics-10-00137-f003:**
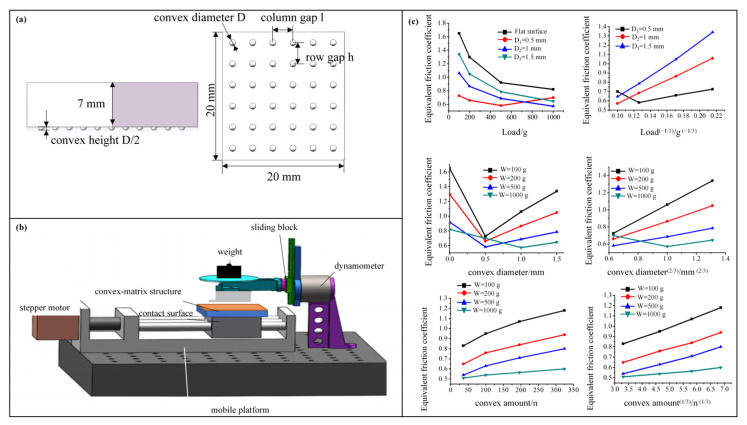
Experimental testing of biomimetic flexible micro-protrusion array friction reduction. (**a**) Biomimetic flexible array sample for experiments, (**b**) friction force testing apparatus, (**c**) variation of equivalent coefficient of friction with load, lattice unit diameter, and number of units.

**Figure 4 biomimetics-10-00137-f004:**
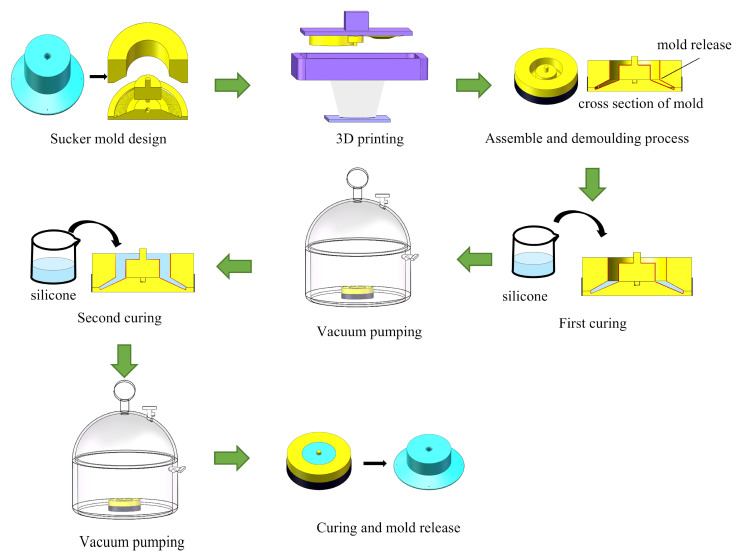
Biomimetic flexible suction cup manufacturing process.

**Figure 5 biomimetics-10-00137-f005:**
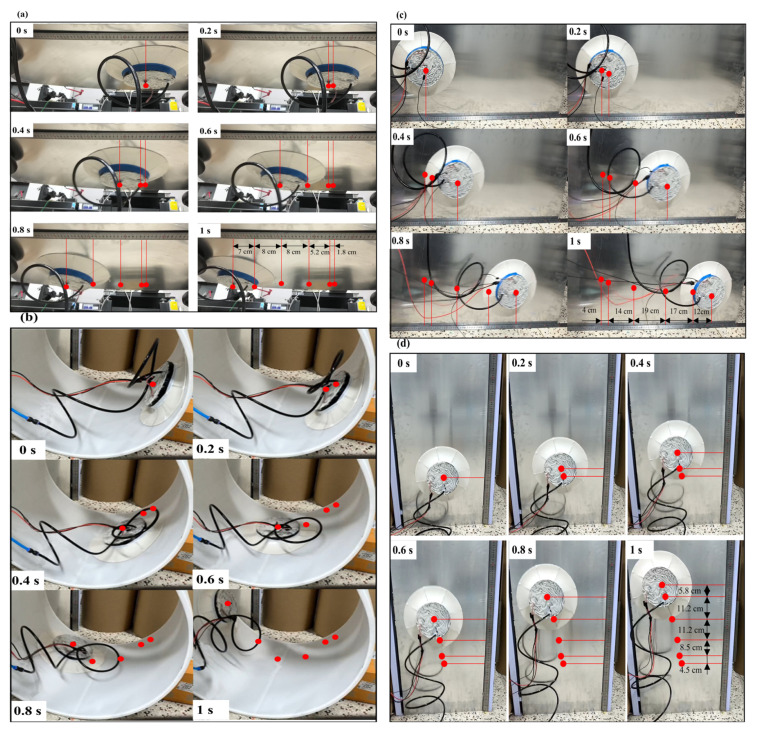
Biomimetic sliding suction cup motion, crawling on (**a**) ceiling plane, (**b**) cylinder inner surface, (**c**) vertical wall surface sideways, and (**d**) vertical wall surface upwards. The red markings indicate the position and distance of the sliding suction cup every 0.2 s.

**Figure 6 biomimetics-10-00137-f006:**
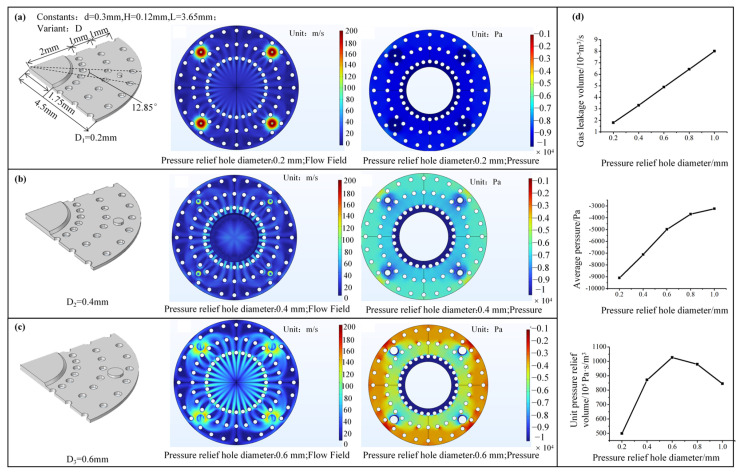
Simulation analysis of bionic relief hole diameter optimization. (**a**) Flow field and pressure distribution of a relief hole with a diameter of 0.2 mm; (**b**) flow field and pressure distribution of a relief hole with a diameter of 0.4 mm; (**c**) flow field and pressure distribution of a relief hole with a diameter of 0.6 mm; (**d**) impact of relief hole diameter on gas leakage, average interface pressure, and unit pressure relief volume. Gary fields represent gas flow in a suction cup.

**Figure 7 biomimetics-10-00137-f007:**
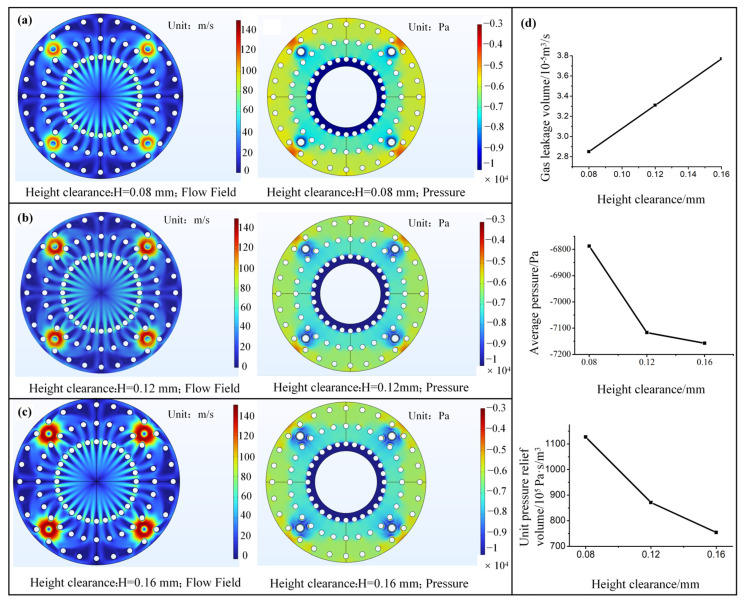
Analysis of biomimetic interface gap height optimization with flow field and pressure distribution of the interface with a height gap of (**a**) 0.08 mm; (**b**) 0.12 mm; (**c**) 0.16 mm. (**d**) Impact of interface gap height on gas leakage rate, average interface pressure, and unit pressure relief.

**Figure 8 biomimetics-10-00137-f008:**
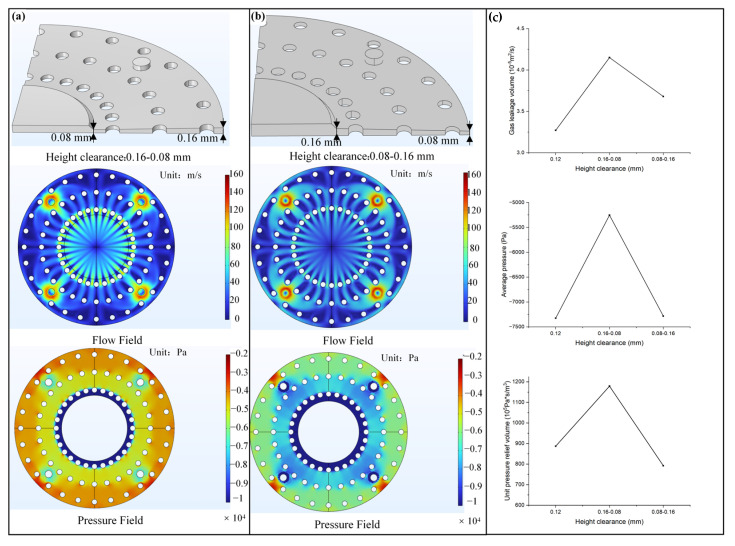
Simulation analysis of biomimetic interface gradient gap with flow field and pressure distribution at the interface gap ranging (**a**) 0.16–0.08 mm; (**b**) 0.08–0.16 mm. (**c**) Effect of interface gap height on gas leakage rate, average interface pressure, and unit pressure volume.

**Figure 9 biomimetics-10-00137-f009:**
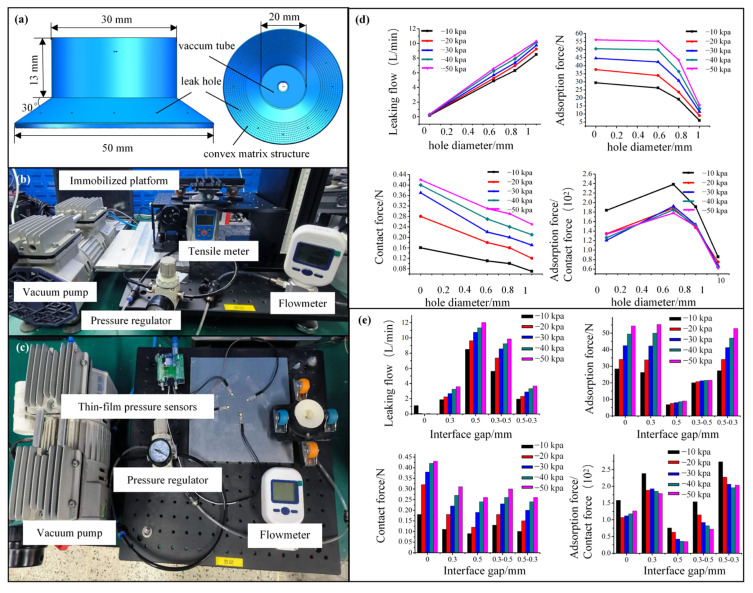
Biomimetic lip edge channel structure for the experimental testing of adhesion force and contact force. (**a**) Design of experimental specimen for biomimetic pressure strength; (**b**) setup of adhesion force testing apparatus; (**c**) setup of contact force testing apparatus; (**d**) impact of vent hole diameter on negative pressure cavity performance; (**e**) impact of interface gap on negative pressure cavity performance, 0.3–0.5 represents height gradually increasing from the center to the edge; 0.5–0.3 represents height gradually decreasing from the center to the edge.

**Figure 10 biomimetics-10-00137-f010:**
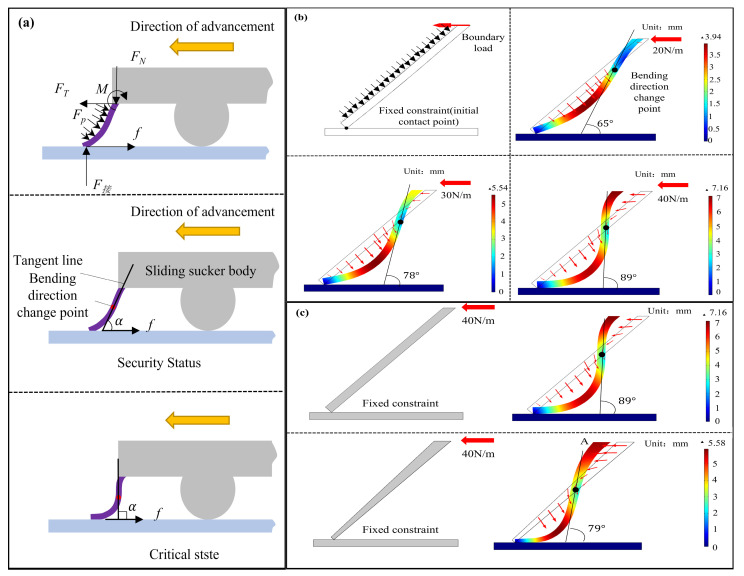
Analysis and optimization of flexible lip edge curling, yellow and red arrow indicate the direction of advancement. (**a**) Force analysis of the critical section of the lip edge during uniform sliding; (**b**) Deformation analysis of the critical section of the lip edge at the front end of the sliding suction cup under different frictional resistances; (**c**) Influence of different cross-section shapes on the deformation of the critical cross-section of the lip edge at the front end of the suction cup.

## Data Availability

The original contributions presented in the study are included in the article, further inquiries can be directed to the corresponding author.

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
