# Peer review of "A Biomimetic Flexible Sliding Suction Cup Suitable for Curved Surfaces"

_biomimetics, 2025, doi:10.3390/biomimetics10030137_

Round 1

Reviewer 1 Report (Previous Reviewer 3)

Comments and Suggestions for Authors

Review of Cui et al.

A biomimetic flexible sliding suction cup suitable for manufacturing on curved surfaces

I remember reviewing the manuscript previously, and in comparison to the previous version, the manuscript has gained much by enlarging the figures and giving additional details on the natural role model and the manufacturing process. However, I am somewhat at a loss why parts of the introduction (which was well written and of interest) has been deleted and replaced by poorly written text. This should be corrected by a native speaker or a more experienced writer.

For example, the sentence:

“Besides, in still current, the pleco can wriggle towards head’s direction without loose its suckermouth entirely, which is quite similar to the movement of sliding suction cup robot.” is not only poorly written but false in term of grammar.

Though not strictly mandatory, I would recommend to add a few supplementary movies that illustrate the movement that can be seen in Figure 5.

Title: The title makes little sense “manufacturing on curved surfaces”. Do you mean “crawling on curved surfaces” or “moving on curves surfaces”? Or, if you simply omit ”for manufacturing”, the sentence reads  “A biomimetic flexible sliding suction cup suitable for curved surfaces”

Line 47: bringing in

Line 55: robots

Lines 82-82: replace “in creatures such as tree frogs and clingfish” by “in tree frogs and clingfishes”

Line 121: why is “Design” at the end of the title with a capital “D”? please write “Deign of biomimetic protrusion structures” or just “Biomimetic protrusions structures”

Line 131: Please indicate what “v” is

Lines 171-173: The figure caption of Figure 3 must be displayed under the figure, not on the next page

Lines 257-373: why was your simulation module so much smaller than the actual suction cup? Can the results for the simulation be well transferred to a larger scale? Please explain.

References:

This is a strange style of referencing that I have never seen before. For example, in [1) and other references, the date reads “2014/03/01. 2014”, so that the year is mentioned twice. Is this the right style of references for the Journal??

There are too many issues (of consistency) in the references to name them all, but for example in some references, the author prenames are mentioned before the family names [22, 23] and the names are written in capital letters, while in others this is not the case. There are often too many spaces or missing spaces. In [22), the co-authors are not mentioned. Some information is missing, for example in [4] and [5] it reads “Editor, ed.^eds., City”, I suspect this a is a place holder for the missing editor(s) and city. In [34], the style of citation of the journal “Scientific reports, 2016, 6(1): 37221.” is radically different from for example [35] “Applied Materials & Interfaces, V. 13, No. 21,2021/06/02. 2021, pp. 24524-31.” Please proof carefully before resubmission!!

Comments on the Quality of English Language

see review report

Author Response

Dear Reviewer:

We want to thank you for your careful and valuable comments.

In the attacted files, we have responsed to your suggestions in a point-by-point manner.

Wish you have a nice day, and happy Chinese New Year.

Reviewer 2 Report (New Reviewer)

Comments and Suggestions for Authors

This paper (“A biomimetic flexible sliding suction cup suitable for manufacturing on curved surfaces”) describes a biomimetic suction robot that utilizes multiple small protrusions on its sealing rim in order to minimize friction while maintaining a suction seal. The authors provide theoretical analysis and empirical analysis to demonstrate the performance of this robot with varying design parameters. The paper is interesting, innovative and informative. My main concern is that there are significant issues with clarity in presenting the technical information. These must be cleaned up.

Bottom p. 2 There are no citations for the mechanism that they describe. If this is the first description for this fish, it needs to be clearer and more rigorous. A labeled diagram would be helpful to demonstrate the actions of the different structures in adsorption-sliding movement.

Line 72: “a vacuum negative pressure” is imprecise terminology. In this situation it is a “reduced pressure” in the mouth cavity. It is not a vacuum, which would be 0 kPa, nor is it a partial vacuum because it is filled with water. It is also not a negative pressure in the strictest sense, which would be below 0 kPa.

Fig 2e needs clearer labeling. The subscripts for the different forces are not showing up clearly, N1, N2 and their arrows should always be aligned properly to each other and the diagram, some arrows in the upper left part of 2e lack labels. All variables should be defined in the caption.

There is a frustrating level of sloppiness in the presentation of the mathematics:

·        Line 127: “the radian of protrusion contact surface” should presumably be “the radius….”

·        Line 133 “elastic module” should be “elastic modulus”

·        Line 131, define v1 and v2

·        Line 134, “radii of Elastic mechanics” I think should be “radii, Elastic mechanics”

·        Line 142, WT should be WT

·        Line 144, AT should be AT

·        Line 155, “linearly proportional to two-thirds of the normal load” – shouldn’t this be to the “2/3 power of normal load”?

·        Line 161 – I could not find Eq (20) or (21). I’m guessing they mean Eq (9) and (10)

The way the authors provide units on the axes of their graphs is not consistent. I prefer units to be in parenthesis after the variable, ie. “Load (g)”. It can also be load/g, as long as it is consistent, but it isn’t consistent in this paper. In Fig 3c, we see “Load W/g”, implying that the units are grams. In contrast, the bottom graph in this figure has the axis label “convex amount n/mm” but in this case mm is not the unit. In this case I assume that it is the number of protrusions per mm so the meaning of the slash has changed completely. Then in Fig. 9 we see units sometimes given in parentheses “Leaking flow (L/min)” and sometimes behind a slash “contact force/g”. Then in the lower right graph of 9d and 9e, we see the slash means that adsorption force is divided by contact force. The authors must be consistent and clear.

In the top two graphs of 3c, the color and symbols used for each of the conditions change, which is confusing. This must be fixed. For example, on the left the 0.5 mm diameter protrusions are represented by red circles, but on the right they are represented by black squares. The same issue is true for the other sizes in these graphs. My first thought on seeing the graph was to wonder how the results could be so different for the black line, but then I realized that the black line represented completely different things in the two graphs.

On Fig.5, the caption should indicate what the red markings represent. It was clear after looking over the figure for a bit, but as a general principle, the markings on the figures need to be as clearly labeled as possible to make it easy for the reader.

Line 267, they should use SI units (kPa) rather than bar. The authors should do this throughout the paper. I also prefer to describe sub-ambient pressures in relative terms “10 kPa below ambient” rather than “-10 kPa”. This distinguishes it from an absolute pressure of -10 kPa (lower than 0 kPa), which is obviously not the pressure in this case.

In Fig.6, The leftmost diagrams will confuse many readers. The drawings on the left imply a solid object with many holes and one protrusion for each sector, which is not the case. I think the diagram on the left represents the airspace under the sucker, but I doubt many readers see it that way because of the way it is drawn. The figure caption must explain what this diagram represents. It makes sense in retrospect, but many people will be confused.

Line 328 “varying sizes of protrusions” – do they vary randomly or systematically? Describe the nature of the variation. From their simulation, it appears that they are assuming a gradient in protrusion height either getting taller moving to the center, or getting shorter as you move to the center.

Line 330 “wedge-shaped height gap” – It would help to define this clearly.

The labels on Fig. 9 (b+c) should be changed so that they clearly identify what they are marking and don’t obscure parts of the apparatus. This is particularly problematic for the thin-film pressure sensors. A good way to do this labeling is with abbreviations that are clearly defined in the caption. The authors should also label the suction cup device in each part of the diagram.

In Fig. 9 d+e why is contact force given in grams, while adsorption force is in Newtons?

In general, the captions should be more informative. For example, “interface gap” in fig 9e should be defined in the caption.  Also, is a gap of 0.3-0.5 referring to 0.3 at the perimeter or 0.3 at the center? For all the figures in the paper, the captions should be sufficiently informative that the reader should be able to read the graphs without having to go back to the text.

Author Response

Dear Reviewer:

We want to thank you for your careful and valuable comments.

In the attacted files, we have responsed to your suggestions in a point-by-point manner.

Wish you have a nice day, and happy Chinese New Year.

This manuscript is a resubmission of an earlier submission. The following is a list of the peer review reports and author responses from that submission.

Round 1

Reviewer 1 Report

Comments and Suggestions for Authors

This paper reported a biomimetic sliding suction cup that is able to climb on various curved surfaces. The essential innovation of this paper might lie in the biomimetic structural design of the suction cup, including a flow channel structure and a matrix of points along the lip edge, to reduce friction aiming at rapid crawling in a continuous adhesive state. The paper suffers from many issues (see details below). Publication is not recommended in the current shape. Rewriting and resubmission are recommended by the reviewer.

 1.     The structure of the manuscript is messy, resulting in poor readability of the paper. It makes no sense to put a highly technically detailed figure at the very beginning of a manuscript. The corresponding author should take the responsibility to restructure the paper.

2.     The quality of the figures in this paper are quite low, which reflects the perfunctory attitude of the authors. For example, in Fig. 3, Fig. 4, Fig. 5, some parts of the board disappear; there is no board in Fig. 7 and Fig. 8 at all; The font sizes in many figures are too small and nor readable.

3. The Method and Material section and the Conclusion section should not written in a bullet form. It looks like a lab report. 

Comments on the Quality of English Language

N.A.

Reviewer 2 Report

Comments and Suggestions for Authors

Dear Authors,

This is an interesting paper on the design and development of a novel suction and robotic motion. I believe this manuscript would benefit from consistent referencing to the biological model. The authors have used various biological terminology ‚pleco, ‚plecoes‚armoured catfish and loach to describe the same biological model mouth features. The biological purpose of the temporary adhesion mechanism is required for readers to grasp the biomimicry element of this manuscript. The difference between a bio-inspired design and a biomimetic one is not clear in this current work.

Line 148. - biological model of ‚pleco‘ needs to be fully described. 

Line 152 - references for the functioning mechanism of the natural system need to be provided.

line 259 - in what way is this a ‚biomimetic flow channel structures, please clearly detail hat is meant by the authors in copying a natural world model for this?

Line 333 - It is my opinion that it would be very interesting to have a reference for or authors could collate and create a graphic describing this variance. The description is not enough to provide this research with biomimetic design steps. Currently this research is only bio-inspired.

Line 360 - are biomimetic vent holes the same features described earlier as biomimetic flow channels?

Figure 7. there are two sets of ‚(d)‘ this is confusing please fix this and clarify in the text.

Line 457 - ‚of friction as the load increases load‘ this is not clear, please fix and clarify.

Reviewer 3 Report

Comments and Suggestions for Authors

Dear editor, dear authors,

This is a very interesting manuscript and I enjoyed reviewing it. I pointed out issues and suggestions than can help to further improve the manuscript.

One major issue, which has to be addressed but can be easily rectified, is that the figures are much too small. When I enlarge them, almost nothing is visible due to the resolution becoming poor. I do not understand why the figures are so minuscule as there is no need to “save space” in an online journal only. Please enlarge the figures considerably so that they have at least twice the size of the current version. I propose to make 4 individual figures out of fig 1 (a, b c, d) or at least to arrange them vertically (one beneath the other) to do the content justice. The figures are well made and highly interesting so that they should also be well visible.

Though not a real issue, the manuscript could much gain by providing videos that illustrate the movements as seen in figure 1.

Page 1, line 25: I don’t understand the term” highly curved motion surface”, is this a highly curved surface in motion?

Page 2, lines 54-55: “friction reduction as the base plate” is it “friction reduction OF the base plate”?

Page2, Material and Methods: why are the figures that describe the manufacturing process in the supplement (only)? I think that they well deserve to be placed in the main text for both better visibility and reference. I think that it would greatly help understanding if figure S2 would be in the main text and appear as the first one, otherwise it is hard to understand what the sliding suction cup system is, especially as the ones depicted in the current figure 1 are so very small.

Lines 121,122: please remove gaps in text. Although most prominent here, this is also an issue in many other parts of the text. Please revise carefully and rectify.

The text format is not consistent: in page 1-3 the text is justified. In contrast, beginning with page 4, the text is not justified.

Line 149: is it “loach” instead of “leach”?

Page 4, Fig.2: Again, why is the figure that small? It must be at least twice the size!

Line 171: “as illustrated in Fig.2(f)”. I cannot find an (f) in Fig.2. I think that you refer to Fig.2(e)

Page 5 and other figures such as 6 and especially 7: please enlarge!

Lines 259-260: I do not understand the statement: “However, introducing biomimetic flow channel structures can compromise the sealing performance of the suction cup.” From what I understood in the abstract , you want to improve the performance of the suction cup by introducing biomimetic flow channel structures(?). So is it just:

However, introducing flow channel structures can compromise the sealing performance of the suction cup.

Or

However, introducing biomimetic flow channel structures can improve the sealing performance of the suction cup.

Fig. 4(d) Bottom:  Perhaps I missed it, but it looks like the “unit pressure relief volume” is not explained or discussed in the text, though this seems to be an interesting result as there is a peak at a relief hole of 0.6 mm. Please explain and discuss or leave away.

Fig. 5(d): Bottom: idem

Fig. 6(c): Bottom: idem

In Fig. 6(c), the values are presented as bar charts. While this is acceptable as such, it is not consistent with the presentation of the values (by connected points) in Fig. 4(d) and 5(d). I suggest that you use one style only. 

Figure 7: there are two sets of subfigures a-d. please change to a-h

Line 428: I cannot follow the sentence part “the shape of the protrusion changes from the dashed to the solid line state”. What is the “dashed line state” and the “solid line state”? Are you referring to a specific figure?

Line 469: please change “ism inimized” to “is minimized”

Lines 495-524, format: the text size is larger than in the rest of the manuscript.

Lines 495-524, content: This is more mere conclusions than a discussion, I strongly suggest that you discuss your results by comparison of relevant literature 

Line 549, Figure S2: what is a “francois”? I know it as a name but not as technical term. Please explain or use other term.

Line 550: I suggest that you delete “manufacturing” from the figure caption so that it reads “Biomimetic sliding suction cup system”.

Comments on the Quality of English Language

Dear editor, dear authors,

This is a very interesting manuscript and I enjoyed reviewing it. I pointed out issues and suggestions than can help to further improve the manuscript.

One major issue, which has to be addressed but can be easily rectified, is that the figures are much too small. When I enlarge them, almost nothing is visible due to the resolution becoming poor. I do not understand why the figures are so minuscule as there is no need to “save space” in an online journal only. Please enlarge the figures considerably so that they have at least twice the size of the current version. I propose to make 4 individual figures out of fig 1 (a, b c, d) or at least to arrange them vertically (one beneath the other) to do the content justice. The figures are well made and highly interesting so that they should also be well visible.

Though not a real issue, the manuscript could much gain by providing videos that illustrate the movements as seen in figure 1.

Page 1, line 25: I don’t understand the term” highly curved motion surface”, is this a highly curved surface in motion?

Page 2, lines 54-55: “friction reduction as the base plate” is it “friction reduction OF the base plate”?

Page2, Material and Methods: why are the figures that describe the manufacturing process in the supplement (only)? I think that they well deserve to be placed in the main text for both better visibility and reference. I think that it would greatly help understanding if figure S2 would be in the main text and appear as the first one, otherwise it is hard to understand what the sliding suction cup system is, especially as the ones depicted in the current figure 1 are so very small.

Lines 121,122: please remove gaps in text. Although most prominent here, this is also an issue in many other parts of the text. Please revise carefully and rectify.

The text format is not consistent: in page 1-3 the text is justified. In contrast, beginning with page 4, the text is not justified.

Line 149: is it “loach” instead of “leach”?

Page 4, Fig.2: Again, why is the figure that small? It must be at least twice the size!

Line 171: “as illustrated in Fig.2(f)”. I cannot find an (f) in Fig.2. I think that you refer to Fig.2(e)

Page 5 and other figures such as 6 and especially 7: please enlarge!

Lines 259-260: I do not understand the statement: “However, introducing biomimetic flow channel structures can compromise the sealing performance of the suction cup.” From what I understood in the abstract , you want to improve the performance of the suction cup by introducing biomimetic flow channel structures(?). So is it just:

However, introducing flow channel structures can compromise the sealing performance of the suction cup.

Or

However, introducing biomimetic flow channel structures can improve the sealing performance of the suction cup.

Fig. 4(d) Bottom:  Perhaps I missed it, but it looks like the “unit pressure relief volume” is not explained or discussed in the text, though this seems to be an interesting result as there is a peak at a relief hole of 0.6 mm. Please explain and discuss or leave away.

Fig. 5(d): Bottom: idem

Fig. 6(c): Bottom: idem

In Fig. 6(c), the values are presented as bar charts. While this is acceptable as such, it is not consistent with the presentation of the values (by connected points) in Fig. 4(d) and 5(d). I suggest that you use one style only. 

Figure 7: there are two sets of subfigures a-d. please change to a-h

Line 428: I cannot follow the sentence part “the shape of the protrusion changes from the dashed to the solid line state”. What is the “dashed line state” and the “solid line state”? Are you referring to a specific figure?

Line 469: please change “ism inimized” to “is minimized”

Lines 495-524, format: the text size is larger than in the rest of the manuscript.

Lines 495-524, content: This is more mere conclusions than a discussion, I strongly suggest that you discuss your results by comparison of relevant literature 

Line 549, Figure S2: what is a “francois”? I know it as a name but not as technical term. Please explain or use other term.

Line 550: I suggest that you delete “manufacturing” from the figure caption so that it reads “Biomimetic sliding suction cup system”.